# Globalization and social distance: Multilevel analysis of attitudes toward immigrants in the European Union

**Ming-Chang Tsai[1], Rueyling Tzeng [2]***

**1** Research Center for Humanities and Social Sciences, Academia Sinica, Taipei, Taiwan, **2** Institute of European and American Studies, Academia Sinica, Taipei, Taiwan

* rtzeng@sinica.edu.tw

**Data Availability Statement:** All relevant data are within the paper and its Supporting information files.

## Abstract

Attitudes toward immigrants can, to a large extent, be determined by certain macro contextual factors. This paper tests a number of proposed hypotheses to illustrate patterns of influence generated by economic and social globalization on perceived social distance relative to immigrants. The European Union (EU) constitutes an ideal study case as its Member States vary in exposure to globalization and attract immigrants from different countries of origin. We conduct a multilevel analysis combining individual level variables from Eurobarometer's recent dataset collected in 2017 and country-level variables from KOF of Globalization Index and other major sources. The results show that individuals in countries with higher degrees of social globalization have lower levels of social distance toward immigrants, while relative level of economic globalization has scant influence. Contact factors are also evaluated for their potential effects. Both casual and close contacts, as specified, reduce social distance. This study contributes to migration studies by offering a clearer specification of how social, rather than economic, globalization interact with contact factors to decrease one's perceived distance from immigrants in the EU.

## Introduction

Globalization continues to enhance contacts and connections between people across societies. In light of increased exposure to, and interaction between, cultural others, a major research issue concerns the volume of legal immigration—how many immigrants should be admitted into a country. In contrast to the issue of *how many* immigrants can be tolerated, *how close or distant* a native-born citizen feels toward immigrants opens new frontiers for migration research. Unlike attitudinal evaluations of a desirable volume of immigration, in this study, we use the social distance concept, defined by Bogardus [1, 2] as the degree of understanding, tolerance, and acceptance that people have towards members of different groups, to better capture how native-born citizens feel when faced with dissimilar others. To use Simmel's metaphor [3], immigrants are not wanderers or visitors who arrive today but leave tomorrow. They are strangers who arrive today and stay tomorrow—they live nearby and may share office space or even become close relatives via marriage. Thus, research questions regarding attitudes

**Funding:** The authors received no specific funding for this work.

**Competing interests:** The authors have declared that no competing interests exist.

toward immigrants concern not merely numbers of immigrants, but contact and interaction, and perceived social distance, especially when, currently, global sympathy toward immigrants seems to be falling as nationalism, xenophobia, and discrimination gain ground in a political climate of protectionism across many wealthy, receiving countries [4].

On the theoretical front, researchers have followed the contact hypothesis [5] to explore how receptivity and positive affect with respect to immigrants evolve out of micro-level social interactions. However, Ceobanu and Escandell [6] urge researchers to consider the influence of macro-level societal interactions, hitherto understudied in the voluminous literature on immigration. This contention is a reminder of a basic premise of contact hypothesis—that an understanding of how an individual interacts with out-group members and develops interpersonal relationships cannot be fully realized without due consideration of the influences generated by contextual factors [5, 7]. Indeed, a country's exposure to the global flux and flow, despite being highlighted by the literature on globalization, has received less attention than it is due given its potential role in facilitating a reduction of social distance between native-born citizens and immigrants [7, 8]. This research aims to fill this gap by providing evidence on how the degree of globalization in each Member State of the European Union (EU), solely or in combination with personal contact factors, influences native-born citizens' attitudes toward immigrants measured by social distance.

Since the second half of the 20th century, Europe has been a target for immigration [9]. Migration became a particularly acute issue after the 2015 European Refugee Crisis, which saw the EU receive more than 1.2 million asylum seekers [10] and more than 4.6 million of long-term (at least a year) immigrants [11]. Although the EU has attempted to develop a common migration and asylum policy, it has never been able to "speak with one voice" due to institutional complexity and internal political dynamics [12]. This probably creates dissimilar attitudes toward immigrants across Member States. Further, as the EU is composed of both high- and low-income countries which have undergone various degrees of globalization (see S1 Table), the EU provides an ideal test case for our proposed hypotheses that globalization contextual factor affects the attitudes of citizens towards immigrants.

We apply multilevel analyses for testing a number of proposed hypotheses. The data at the individual level are drawn from Eurobarometer 88.2 (October 2017), which provides a comprehensive measure of social distance in explaining to what extent a feeling of closeness with immigrants can be observed through cross-national comparisons within the EU. The internal migration of EU citizens between Member States is not covered in this dataset as its questionnaire clearly defines immigrants as "people born outside the European Union, who have moved away from their country of birth and are at the moment staying legally in (OUR COUNTRY)." At the country level, the main data is from KOF of Globalization Index (KOFGI). As each force in the various dimensions of globalization does not necessarily co-vary, we distinguish between the economic and social dimensions. Moreover, in the models, we also control for the most cited variables in the research on attitudes toward immigrants, the unemployment rate, and degree of right-wing populism. We find that although contact with immigrants remains a strong predictor of acceptance of immigration, individuals in countries with greater degrees of social globalization further reduce their social distance with respect to immigrants while a country's economic globalization plays little role.

## Globalization and contact: Macro- and cross-level hypotheses

Globalization consists of diverse forces simultaneously generating wanted and unwanted consequences. Economic globalization, arguably the most powerful facet of globalization, channels flows of capital, goods and services across international boundaries. Many scholars find that

global economic flows generate favorable effects for national growth, employment and income [13, 14]. Thus foreign goods and services are increasingly available at low prices, additional export opportunities for domestic enterprises arise, and there are more employment opportunities thanks to new businesses foreign and domestic. On the other hand, economic globalization's adverse impacts have been highlighted, as well: inequality of income distribution, job insecurities, environmental pollution, and so on [15]. While economic globalization has lifted millions out of poverty across many regions, for many people in the global north [16], eroding income and precarious working conditions seem to have emerged as the "new normal".

The evidence for economic globalization's influence on receptiveness to increased immigration is mixed. Koster [17] finds that economic engagement with other countries contributes to individuals' willingness to assist immigrants. In contrast, Kaya and Karakoç [8] report that a country's openness to trade is correlated with anti-immigrant sentiments (measured by acceptance of immigrants as neighbors) among the mass public; they also found that volumes of foreign direct investment had no effect. These results seem to reflect Guillén's [18] summary of three conflicting camps of globalization as civilizing (integrative), destructive, or feeble. Neoliberals argue that globalization functions to civilize the globe, an argument that stands in stark contrast to the position of critics who see globalization as a destructive force that only harms the domestic economies of which governments are unable to protect their populations. The third camp is ambivalent in that it sees globalization—although represented by a spectacular mobility of finance, technology, and manufacturing—as a feeble process that has not yet challenged the fundamental features of modern society, nor reshaped basic popular value positions in its favor.

Beside increasing capital flows and the transnational division of labor, globalization processes also entail rapid expansion of social ties and the broad diffusion of ideas at scale across regions. Indeed, social globalization, as such, is indicative of increasing the number of people engaged in making contacts, communicating, and engaging in exchanges on various platforms —physically or via the Internet—on a global scale. Interactions via online social media, or in person by way of travel for leisure or work, have been found to enhance a cosmopolitan disposition, which favors universalist viewpoints and increases openness toward cultural others. Szerszynski and Urry [19] describe cosmopolitanism by juxtaposing two interesting ideal types: the map-reader, and the way-finder. What globalization has facilitated is a way of seeing the world with an abstracted visual approach such that the map is read by means of comparing and contrasting one place with others as imaged from afar. This mindset differs from a local perspective that tends to read a map as a means of finding a way from one shop to another. Exposure to ideas, information and images from outside one's locale has led to the adoption of transnational identities, that is, a belief that an individual is a "world citizen" rather than limited to a particular national identity [20, 21]. This does not mean forsaking loyalty to the vernacular culture into which an individual is born, for the issue is not whether tension or irreconcilability between the two exists. Rather, it is an awareness of interdependence, mutual influence, and a consciousness of the world as a whole that is highlighted in the global mapping [22, 23]. Social globalization does not necessarily bring mutual understanding, but may lead to cultural conflict and xenophobia. When the numbers of imported cultural goods or immigrants exceed a certain level, natives may fear a cultural invasion, which threatens their traditions (values and customs) and ways of life. This often stimulates inter-group violence as social groups attempt to reestablish and reaffirm their conventions and sense of identity [24]. Much research has shown how locals react towards outsiders in a globalized world, but few examine how the characteristics of global exposure in a country influences the mass publics' attitudes toward others. Koster [17] finds that a country's openness to social globalization positively influences its citizens' willingness to help immigrants, while Salamońska [25] shows this

did not play any role in shaping EU citizens' attitudes toward immigrants from other EU countries.

In this section, we introduce the contact factor and its interaction with globalization to enrich our theoretical explanation. Originally, contact theory posits a general hypothesis that prejudice toward outgroups such as immigrants is reduced by interpersonal contact [5]. Most research elaborates the *process* of reduction of prejudice by arguing that contacts, especially repeated contacts, lead to a reduction of bias, as in this way knowledge about the out-group is enhanced, cognitive dissonance resolved, and affective ties generated. If a judgmental evaluation occurs after contact, it is more likely to be a 'person-based' processing of information than 'category-based' speculation [26]. When the number and quality of contacts increases, immigrants are seen as complex, variable, and individual, and therefore each is deemed different rather than lumped into a collective, unwelcome, outgroup category. This decategorization discourages negative stereotyping. That is, group salience (or group membership) is downplayed, discomfort reduced, and greater trust can be developed towards immigrants.

The contact hypothesis, however, does not offer sufficient discussion of *how* or *why* positive interpersonal relationships might emerge, and even become quite common between the native-born and immigrants [27]. Wright, Brody and Aron [28] advance an explanatory self-expansion theory. People, in general, have a basic interest in enhancing their personal efficacy by acquiring new resources, perspectives, and experiences that facilitate the achievement of present or future goals [29]. Opportunities to interact with dissimilar others, or cross-group members, offer the potential for gaining access to these "goods" and enhance self-efficacy [30, 31]. Through this mechanism of self-expansion, the outgroup is included in a person's self as a new close relationship, although this person is aware that he or she is not (or cannot be) included in the outgroup. This theory is helpful in understanding why bias reduction does not follow intergroup contacts afterwards. This is not to deny that as a feeling of interpersonal closeness evolves alongside contacts, there is a possibility of rejection by one's ingroup and a consequent loss of the self. This counterpoise (self-consistency) predicts a weaker effect produced by contacts [29].

While contact theory is optimistic about the influence of intergroup contact in reducing prejudice towards immigrants, one of its major theoretical contribution lies in the specification of certain contexts in which contact effects are most likely to be realized. Indeed, in some situations, for instance, during a competitive environment for resource distribution, those with more contacts seem to have incurred more conflicts rather than greater peace, which prevents what self-expansion theory has expected about the potential of friendship to happen. Allport yet contends that prejudice "may be reduced by equal status contact between majority and minority groups in the pursuit of common goals. The effect is greatly enhanced if this contact is sanctioned by institutional supports (i.e., by law, custom, or local atmosphere), and provided it is of a sort that leads to the perception of common interests and common humanity between members of the two groups" [5]. Pettigrew [32] marks four key conditions for the positive effects of intergroup contact to occur: 1) having equal status, 2) sharing common goals, 3) engaging in intergroup cooperation, and 4) benefiting from support of authorities, law or custom. Note that these are not necessary causes for prejudice reduction. Rather, they are moderating factors which are needed for the contact factor to generate positive outcomes. We agree that the specification of the environmental cues for contacts to lower prejudice are highly important in testing the contact hypothesis [33]. In methodological terms, this suggests that with a specific contextual (moderating) factor in models, it is more likely to detect the contact factor's favorable effects.

In light of this facilitating context argument, we propose that a country's global exposure can provide a favorable environment. It is not listed in conventional contact arguments, as

attention has focused on the relatively *immediate* contexts in which ingroups interact with outsiders, for instance, in classrooms, neighborhoods or workplaces either in experimental designs or observational studies [34–36]. Our interest in globalization leads to a necessary turning of attention to a country's openness toward the global system. As globalization advances deeper in a country, it is likely that policy reforms will have been implemented such that the economy will be increasingly open to trade and foreign capital and such that the society is more receptive to cultural exchange across borders and immigration. That is, in an environment in which people have more experience with foreign investment, technology, ideas, culture, and more transnational relationships due to their work or leisure, they are more inclined to see immigration as a usual phenomenon, not a form of culture shock, or a source of competition in the labor market or the struggle for public resources via welfare programs. Immigration can be even considered a contributor to economic growth and cultural diversity for the destination societies. In contrast, in a largely closed country with little previous experience accommodating global forces, more contacts could result in tensions and hostility towards outsiders because the native-born feel distant and aloof with respect to immigrants and have little interest in knowing or associating with them. Situated in a low global exposure environment and lacking common interests, little positive is likely to result from contacts between the native-born and immigrants.

In sum, this study proposes four major hypotheses for testing. At the macro-level theorizing:

H1: The more globalized a country's *economy* is, the more the people feel receptive to immigrants.

H2: The more globalized a country's *society* is, the more the people feel receptive to immigrants.

As our theoretical argument highlights globalization as important macro-level environmental cues (facilitating contexts) for the contact factor at the micro level, the following hypotheses involves *cross*-level interactions:

H3: Contact with immigrants reduces social distance toward immigrants more substantially when happening in a country with a greater level of *economic* globalization.

H4: Contact with immigrants reduces social distance toward immigrants more substantially when it happens in a country with a greater level of *social* globalization.

Investigating how macro- and cross-level factors affect feelings toward immigrants, we consider a number of important covariates which have been emphasized in the literature, such as demographic characteristics and political ideology. Simultaneously considering multiple factors enhances the reliability of the findings presented in this study.

## Materials and methods

Individual-level data are drawn from Eurobarometer (EB)—a series of public opinion surveys covering a wide range of topics conducted since 1974, originally on behalf of the European Communities and then for its successor, the EU. For this study, we draw on Eurobarometer 88.2 (October 2017): Integration of Immigrants in the European Union and Corruption. This survey covers a comprehensive set of questions about immigration allowing for a better measure of perceived social distance. Other surveys are less complete and have thus been passed over. For example, the European Social Survey (2014, the latest version with the questions on social distance) asks whether respondents would mind having an immigrant as their boss, or a

close relative via marriage; the World Value Survey (Wave 7 for 2017–2022) merely asked respondents whether or not they would like to have immigrants as neighbors.

This EB dataset has a sample size of 28,080 and was based on a multistage probability sampling. The field method was face-to-face interview with a CAPI (Computer Assisted Personal Interview) system. Respondents were aged 15 years and over, and resident in any of the 28 Member States of the EU. Although UK voted to exit EU on June, 2016, 'Brexit' did not come into effect until January 31, 2020. Therefore, UK was included in this dataset. Approximately one thousand responses per country were collected, except for the UK (1,382 samples), Germany (1,554), and three smaller countries: Cyprus, Luxembourg and Malta (approximately 500 each). As Malta is a very small country, with a population of less than 500,000 in 2017, it is not included in our analysis. Also excluded were respondents who were not EU citizens. While there were 27,462 eligible cases, 26,223 were used in this analysis after excluding a small percentage (4.5%) with missing information.

Social distance in Bogardus's [2] survey research was originally measured by asking respondents to report whether they would admit members of an out-group in a variety of ways, ranging from the least to the most social distance in the following categories: close kinship by marriage, personal chums, neighbors, workers, citizenship, visitor, and exclusion from their country. The EB adopted a similar design. The respondents were asked how comfortable or uncomfortable they would feel, personally, in each of the six kinds of social relations with an immigrant: as their manager, work colleague, neighbor, doctor, family member (including partner) and friend. The answers for each question were coded as 1 = totally comfortable, 2 = somewhat comfortable, 3 = somewhat uncomfortable, 4 = totally uncomfortable. We first conducted a factor (principle component) analysis to check the level of inter-item correlations. The result showed that these six items converge onto one main factor, suggesting together they represent one underlying cause. The Cronbach alpha coefficient (reliability) for these six items is 0.95, indicating that the items have a very high internal consistency. Therefore, we used the mean value of the sum of the six items as an index for social distance (range from 1 to 4). The higher the score, the greater social distance a respondent expresses relative to immigrants.

In measuring a country's level of globalization, we use the KOFGI—a composite index designed by Dreher [13]. This taps three dimensions of globalization, with each dimension covering *de facto* and *de jure* aspects—the former measures actual flows and activities and the latter, policies, resources, conditions and institutions that enable or foster actual flows and activities [37]. The index of each dimension is measured on a scale from 1 (least) to 100 (most globalized). In this study, both economic and social aspects of globalization are evaluated according to our arguments and the proposed hypotheses.

Economic globalization is measured based on a country's performance in trade and finance. The original measure is constituted of eight items, including trade, stocks of foreign direct investment, trade barriers, capital restrictions, taxes on international trade, etc. [38]. Social globalization is composed of indexes of interpersonal, informational and cultural exchanges and is measured by a total of eleven items, including telephone traffic, internet users, television access, trade in newspapers, international tourism, foreign population, number of McDonald's restaurants and Ikea furniture stores, trade in books, etc. Some aspects of the social globalization index may appear problematic, such as the number of McDonald's and IKEA stores, but it is a way to measure the ability to understand and accept foreign cultural values [37]. In the absence of other comprehensive measures of globalization, the KOFGI is the best index of social and economic globalization [39, 40]. An index of political globalization, indicating level of involvement in intergovernmental organizations by a country, is also provided by the KOFGI dataset. We do not use it in this study as there are few theoretical supports for its potential influence on social distance. Our preliminary analysis also shows a thin and non-

significant correlation between social distance and political globalization. To avoid fluctuation, in this research we used the mean index of three years, 2015–17, for economic and social globalization, respectively. The country-level variables for EU members are displayed in the S1 Table.

Casual contact and close (acquaintance) contact with immigrants are separately measured in our design. This is necessary because superficial contact with out-group members often confirm stereotypes, while acquaintance contacts are more likely to provide opportunities to learn about out-group members, sustain a relationship, and thus lessen prejudice and hostility [5]. Casual contact is measured by response to six questions in the EB survey. Respondents were asked how often they interacted with immigrants on six occasions: at work, in a childcare center, school or university, in the neighborhood (e.g. shops, restaurants, parks and streets), while using public services (e.g. hospitals, local authorities' services, public transport), during sport, volunteer or cultural activities, and when using household services (e.g. home helps, cleaners, repair technicians or babysitters). Their responses were coded as 0 = less often or never, 1 = at least once a year, 2 = at least once a month, 2.5 = don't know, 3 = at least once a week, 4 = daily. We calculated the mean for the sum of the six questions (value range is between 0 and 4); a higher score indicates more casual contact.

The EB survey asks the following question, which we used to measure respondent's close contact with immigrants: whether they (1) have "friends who are immigrants currently living in (OUR COUNTRY)" (coded 1), (2) have "family members who are immigrants currently living in (OUR COUNTRY)" (coded 2), and (3) have "both friends and family members who are immigrants currently living in (OUR COUNTRY)" (coded 3). Those who have neither relationship were coded 0. Higher scores indicated more close contact.

Variables for additional consideration concern basic demographic background: gender, age, marital status, residence area, education, occupation and class position (see Table 1 for the detailed groupings for each variable). Since the EB survey does not collect information on income, we use the question on level of difficulty to pay bills as a proxy. We also look at political ideology. It has been argued that individuals who perceive immigrants as posing economic and cultural threats tend to be unwelcoming due to the increased competition in the job market, for welfare benefits, housing, and other resources, and due to perceived challenges of the traditions, group boundaries and identity [41, 42]. Individuals with right-wing political orientations tend to hold sentiments opposing immigrants due to perceived economic and cultural threats [43, 44].

A country's unemployment rate and the prevalence of right-wing populism are two societal factors frequently investigated in studies of attitudes toward immigrants [43, 44]. Indeed, support for right-wing populist parties seems to have grown significantly all across Europe in the wake of the 2015 refugee crisis [45]. It also signifies a surge in anti-immigration sentiment [46]. Unemployment rates are measured in terms of the number of unemployed as a percentage of total labor force, and calculated by the average value for the years 2015–2017 from the World Development Indicators overseen by the World Bank. To indicate the prevalence of right-wing populism in a society, we use the data from the Timbro Authoritarian Populism Index (TAP) which contains information on populist parties' electoral results in national parliamentary elections since 1980 in many democratic European countries. Mudde [47, 48] identifies a party as populist if it portrays society as separated into homogeneous and antagonistic groups, speaks of 'the pure people' versus 'the corrupt elite', or maintains that politics should be an expression of the *volonté générale* (general will) of the people. Among various types of populist parties, right-wing parties emphasize nationalism in contrast to the socialism promoted by left-wing parties [49]. The TAP dataset defines parties promoting radical nationalistic ideologies and using xenophobic rhetoric as right-wing populist ones [45]. In this research,

**Table 1. Summary statistics.**

| Variables | Percentages | Means (SD) |
|---|---|---|
| *Individual level* (N = 26,223) | | |
| Social distances | | 1.94 (0.84) |
| Gender | | |
| Female | 55.71 | |
| Male | 44.29 | |
| Age | | 51.80 (17.86) |
| Marital status | | |
| Married/cohabited | 65.35 | |
| Single/divorced/separated/widow | 34.65 | |
| Residence area | | |
| Rural/village | 31.64 | |
| Small/middle town | 40.91 | |
| Large town | 27.45 | |
| Education | | |
| Lower secondary or less | 15.52 | |
| Upper secondary | 46.68 | |
| Tertiary | 37.80 | |
| Occupation | | |
| Managers/professionals | 12.55 | |
| Self-employed | 5.16 | |
| Lower white collar | 17.23 | |
| Manual workers | 16.51 | |
| House work | 4.81 | |
| Students | 5.03 | |
| Temporarily not working | 5.40 | |
| Retired | 33.31 | |
| Difficulties to pay bills | | |
| Most of the time | 9.35 | |
| From time to time | 25.42 | |
| Never | 65.23 | |
| Class | | |
| Working/lower middle class | 41.74 | |
| Middle class | 47.00 | |
| Upper/higher middle class | 7.66 | |
| Other/none/refusal/don't know | 3.60 | |
| Political ideology | | |
| Right | 31.03 | |
| Left | 49.95 | |
| Refusal/don't know | 19.02 | |
| Casual contact | | 1.06 (1.02) |
| Close contact | | 0.53 (0.87) |
| *Country level* (N = 27) | | |
| Economic globalization index | | 80.20 (5.28) |
| Social globalization index | | 84.52 (3.95) |
| Unemployment rate | | 8.61 (4.39) |
| Right-wing populism | | 14.15 (13.97) |

Standard errors for continuous variables in parentheses.

the prevalence of right-wing populism is measured by the level of support for right-wing populist parties among voters in each EU country. We used the average value of election results in terms of percentage of votes cast for right-wing populist parties between 2015–2017 from the TAP dataset. High support for populist parties in a country reveals a social climate that is more nativist, xenophobic, and anti-cosmopolitan, which leads to greater prevalence of social distance attitudes toward immigrants.

Note that we did not include the size of a country's immigrant population because our preliminary analyses did not find this to be a significant influence on social distance attitudes. Moreover, the index of social globalization already covers this variable, as mentioned above. Therefore, it would be redundant to include the number of a country's immigrants in our study.

Estimation. This study pooled respondents from a diverse set of populations for evaluating cross-country difference in social distance attitudes. As the data involve both individual and national units, conventional least squares techniques, based on an assumption of independent observations from a homogeneous sub-population, can be inadequate as the estimation may perform inefficiently given correlated observations within the same country or "cluster". Ignorance of cluster-specific effects can lead to smaller standard errors and "spuriously significant" results [50]. We therefore decided to use the hierarchical linear models to estimate individual and contextual effects simultaneously with a weighted sample by taking each country's population size into consideration. An additional advantage is that multilevel modeling provides flexible specification of individual- and country-level information and the error structures at both levels. Specifically, it takes into account the differences across countries and provides information on random intercept variance so that the extent to which countries differ in social distance attitudes can be estimated (that is, assuming heterogeneity in the intercepts and slopes).

More importantly, this technique allows flexible modeling of the cross-level interaction effects for testing H3 and H4—that is, whether contacts are more likely to generate favorable influences in countries with higher levels of economic and social globalization. These techniques become standard practice in the context of pooled cross-national datasets [51, 52]. Cross-level interactions, as such, are often examined for their potential influence in the multilevel analysis. In light of this interest, our estimation considers the interaction effects of country-level indicators and the two individual contact behaviors. In calculating the interaction terms, the elements were centered beforehand in order to more easily interpret the results while not affecting the macro structures (the correlation matrix or modeling fit) in estimation [53].

## Results

Table 2 presents the correlation coefficients for the social distance variable, and four independent variables at the country level, used in this study. Although economic and social

**Table 2. Correlations among the dependent variable and country level variables (N = 27).**

|  | Social distance | Economic globalization index | Social globalization index | Unemployment rate |
|---|---|---|---|---|
| Economic globalization index | -0.388* |  |  |  |
| Social globalization index | -0.673*** | 0.654*** |  |  |
| Unemployment rate | -0.012 | -0.438* | -0.243 |  |
| Right-wing populism | 0.456* | -0.030 | -0.202 | -0.293 |

*** p < .001,
** p < .01,
* p < .05

globalization are positively correlated (r = .654, p < .001), social globalization is negatively correlated with social distance (*r* = -.673, *p* < .001) but economic globalization carries a much smaller negative coefficient at border-line significance level (p = .0456). These suggest that each aspect of globalization may differently exert influence. Furthermore, right-wing populism is positively correlated with social distance (*r* = .456, p = .0169) while a country's unemployment rate holds little correlation. Note that we have checked whether high correlation among the predictors used is a potential threat to stability in the estimation. We found that the variance inflation factors behave well (all under 4) and concluded that multicollinearity is not an issue.

Although we are interested in how the country context and its interaction with the individual contact behaviors influence individual social distance relative to immigrants, examining how country contexts affect individual contacts will enrich our understanding of the importance of the macro structure. Column 1 of Table 3 of the multilevel analysis shows that economic globalization does not seem to affect causal contact, whereas social globalization has a positive influence—that is, a country immersed in wider social and cultural exchanges (in contrast to increased foreign trade or capital investment) is more likely to see that such casual contacts are popular among the mass public. The intra-class correlation coefficients are high at .18, indicating large variation across countries and justifying the use of hierarchical linear models. In the second model on Table 3, social globalization consistently generates favorable influence by boosting more close relationships, whereas economic globalization does not. In sum, a country's greater degree of social globalization increases its citizens' casual and close contacts with immigrants, but economic globalization plays little role.

Turning now to the testing results of the four major hypotheses, we first estimate the effects of predictors both at the country and individual level in Model 1 of Table 4, then add a cross-level interaction term to the equations. Given the large sample size of respondents in the analysis, our outcomes are quite stable and thus reliable with regard to the coefficients from the interaction term and their original input factors. Therefore, we only present the results with a significant interaction term as Model 2 of Table 4.

In Model 1, both types of contact decrease social distance as expected, even when controlling other important variables at the individual level: age, education, occupation, income (difficulty to pay bills) and political ideology. Of the two control variables at the country level, the prevalence of right-wing populism increases social distance, whereas the unemployment rate plays little role despite immigration being considered a potential threat and source of competition in the labor market. As to the two main independent variables at the country level, social globalization substantially reduces social distance (supporting H2), while economic globalization has little influence (rejecting H1). In other words, EU citizens in a country with a higher degree of social globalization tend to have more favorable attitudes toward immigrants, an effect not found in EU countries where the economy is highly globalized.

It is expected that in a context of deepened globalization, contact significantly decreases social distance. Alternatively, in a closed environment, contact, as measured here, would bring about distrust rather than generate sympathy or mutual understanding. To test these hypotheses, we generate four cross-level interaction terms (two globalizations time two contacts). Model 2 on Table 4 shows the only significant interaction, that is, the positive result of close contact × social globalization (*b* = .009, p < .05). These outcomes appear to refute H3 and H4, both of which expect a negative sign (that is, social distance is further reduced). To interpret the result more clearly, we present it graphically in Fig 1, in which social globalization is a moderating variable for close contact as a predictor of social distance (all three variables are centered in model) [53]. The slope of close contact decreases somewhat given a high degree of social globalization when compared to countries with a medium or low degree of social

**Table 3. Multilevel analysis of contacts (N = 26,223/27).**

|  | Casual contact | Close contact |
|---|---|---|
| *Individual level* |  |  |
| Gender (Female = 0) | -0.020 (0.013) | -0.016 (0.012) |
| Age | -0.009*** (0.001) | -0.005*** (0.001) |
| Marital status (Married/cohabited = 0) | -0.047** (0.015) | -0.052*** (0.014) |
| Residence area (Rural/village = 0) |  |  |
| Small/middle town | 0.140*** (0.034) | 0.040 (0.025) |
| Large town | 0.373*** (0.058) | 0.102** (0.033) |
| Education (Lower secondary or less = 0) |  |  |
| Upper secondary | 0.100** (0.036) | 0.051* (0.025) |
| Tertiary | 0.158*** (0.033) | 0.124*** (0.027) |
| Occupation (Managers/professionals = 0) |  |  |
| Self-employed | -0.135** (0.043) | 0.019 (0.030) |
| Lower white collar | -0.171*** (0.046) | -0.036 (0.024) |
| Manual workers | -0.242*** (0.058) | -0.041 (0.025) |
| House work | -0.451*** (0.055) | -0.098 (0.052) |
| Students | -0.138** (0.048) | 0.042 (0.042) |
| Temporarily not working | -0.409*** (0.054) | -0.101** (0.038) |
| Retired | -0.399*** (0.054) | -0.053 (0.030) |
| Difficulties to pay bills (Most of the time = 0) |  |  |
| From time to time | 0.006 (0.051) | -0.010 (0.034) |
| Never | -0.106 (0.057) | -0.077* (0.031) |
| Class (Working/lower middle class = 0)[a] |  |  |
| Middle class | -0.004 (0.031) | 0.000 (0.018) |
| Upper/higher middle class | 0.051 (0.041) | 0.024 (0.032) |
| Leftist ideology (Right = 0)[a] | 0.055 (0.029) | 0.072** (0.026) |
| *Country level* |  |  |
| Economic globalization index | -0.012 (0.018) | -0.001 (0.007) |
| Social globalization index | 0.078*** (0.017) | 0.049*** (0.010) |
| Unemployment rate | 0.035*** (0.009) | 0.011 (0.009) |
| Right-wing populism | -0.003 (0.004) | -0.002 (0.002) |
| Constant | -4.352 (1.229) | -3.389 (0.538) |
| Intraclass correlation coefficient | 0.182 | 0.077 |

*** p < .001,

** p < .01,

* p < .05

[a] Respondents answered "Don't know," "Refusal," "Others," etc. are not presented in the table.

globalization. Here, what is referred to as high and low is defined by one deviation above the mean (the medium). Given a small (albeit statistically significant) practical influence of globalization's moderation, the main result remains that close contact helps alleviate the feeling of low attachment to immigrants across EU countries with varied levels of social globalization.

## Discussion and conclusion

Previous research on attitudes toward immigrants has focused mainly on the effects of individual characteristics, with few studies taking the macro structural configurations into consideration. Among the macro level factors, the prevalence of right-wing populism, unemployment

**Table 4. Multilevel analysis of the social distance (N = 26,223/27).**

|  | Model 1 | Model 2 |
|---|---|---|
| *Individual level* |  |  |
| Gender (Female = 0) | 0.017 (0.015) | 0.017 (0.015) |
| Age | 0.002*** (0.001) | 0.003*** (0.001) |
| Marital status (Married/cohabited = 0) | 0.005 (0.014) | 0.006 (0.014) |
| Residence area (Rural/village = 0) |  |  |
| Small/middle town | 0.002 (0.020) | 0.002 (0.020) |
| Large town | -0.003 (0.031) | -0.004 (0.030) |
| Education (Lower secondary or less = 0) |  |  |
| Upper secondary | -0.061** (0.021) | -0.058** (0.021) |
| Tertiary | -0.155*** (0.024) | -0.152*** (0.023) |
| Occupation (Managers/professionals = 0) |  |  |
| Self-employed | 0.013 (0.042) | 0.014 (0.042) |
| Lower white collar | 0.066** (0.025) | 0.066** (0.024) |
| Manual workers | 0.115*** (0.025) | 0.116*** (0.025) |
| House work | 0.087** (0.027) | 0.086** (0.026) |
| Students | 0.014 (0.036) | 0.014 (0.035) |
| Temporarily not working | 0.032 (0.035) | 0.031 (0.035) |
| Retired | 0.055 (0.031) | 0.055 (0.031) |
| Difficulties to pay bills (Most of the time = 0) |  |  |
| From time to time | -0.043 (0.031) | -0.043 (0.031) |
| Never | -0.207*** (0.029) | -0.207*** (0.029) |
| Class (Working/lower middle class = 0)[a] |  |  |
| Middle class | 0.009 (0.026) | 0.010 (0.026) |
| Upper/higher middle class | 0.026 (0.037) | 0.027 (0.038) |
| Leftist ideology (Right = 0)[a] | -0.148*** (0.030) | -0.149*** (0.029) |
| Casual contact | -0.103*** (0.012) | -0.103*** (0.012) |
| Close contact | -0.183*** (0.016) | -0.194*** (0.017) |
| *Country level* |  |  |
| Economic globalization index | 0.001 (0.013) | 0.001 (0.013) |
| Social globalization index | -0.037* (0.016) | -0.037* (0.016) |
| Unemployment rate | -0.003 (0.008) | -0.002 (0.009) |
| Right-wing populism | 0.008** (0.003) | 0.008** (0.003) |
| *Interaction terms* |  |  |
| Close contact*Social globalization index |  | 0.009* (0.004) |
| Constant | 5.278 (0.904) | 5.219 (0.868) |
| Intraclass correlation coefficient | 0.215 |  |

*** p < .001,

** p < .01,

* p < .05

[a] Respondents answered "Don't know," "Refusal," "Others," etc. are not presented in the table.

rates, and size of immigrant populations have been investigated. In light of this, our study examines the influence of a country's degree of globalization by proposing a series of hypotheses for testing against EU population surveys. Adapting an approach of the multidimensional and dynamic force of globalization, we provide empirical evidence revealing how two major aspects of globalization, economic and cultural, differently influence citizens' social distance

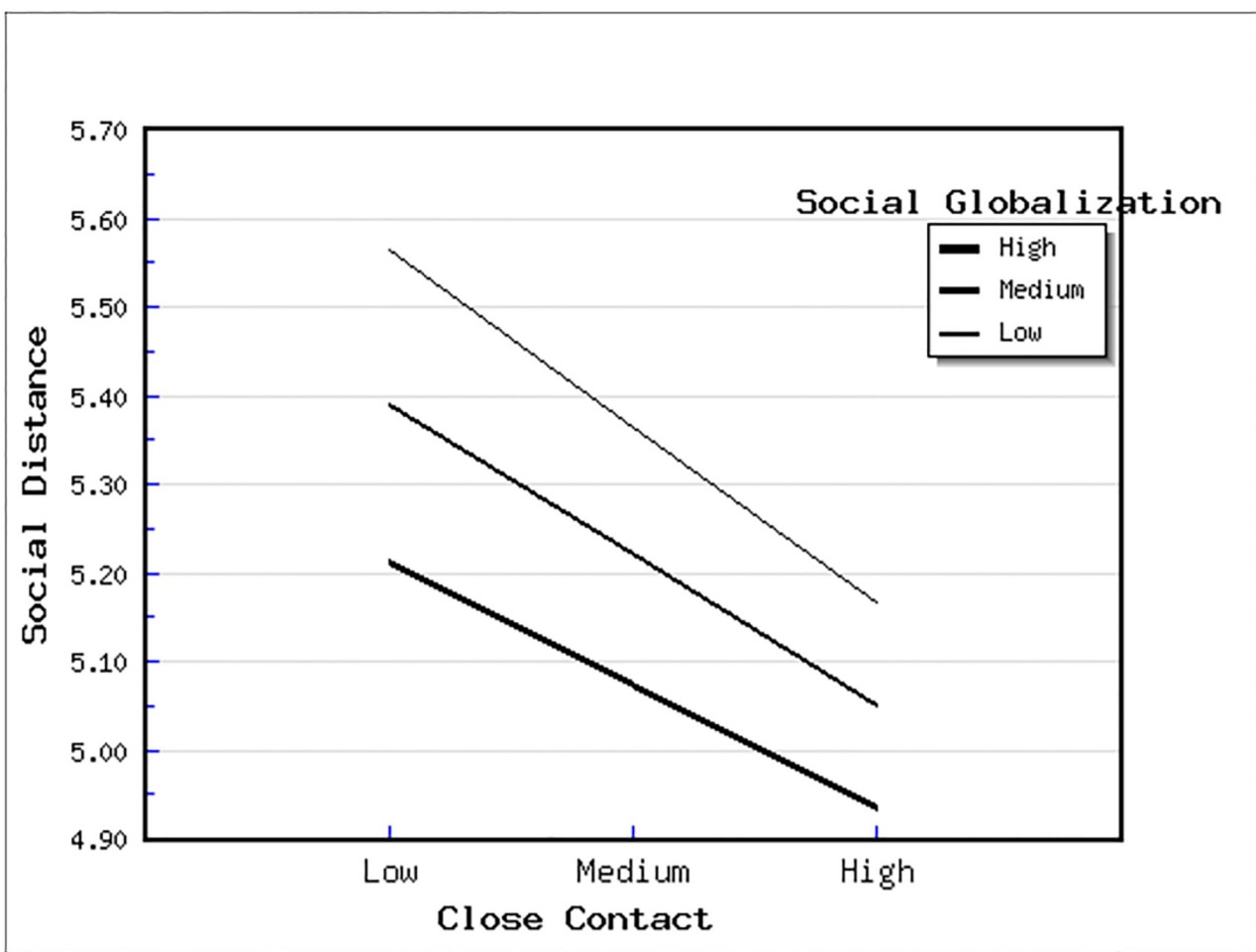

**Fig 1. The interaction effect of close contact and globalization on social distance.**

with immigrants. One main result obtained indicates that a country's higher level of social globalization both enhances individuals' contacts with immigrants and boosts a feeling of closeness to, and acceptance of immigrants. This result is consistent with Koster's [17] study.

While globalization has been deepening worldwide, a backlash has also arisen, insofar as trade liberalization has fallen, foreign investment plateaued, protectionist policies strengthened, and political parties grown increasingly opposed to globalization in richer countries [54–56]. Economic globalization seems to have also stimulated the rise of protectionism and support for right-wing populism in many Western countries [4]. Nevertheless, these consequences are mostly due to institutional failures in response, yet immigrants are often blamed for increased economic adversity and intergroup conflicts. Our research finds that the prevalence of right-wing populism in a country increases individuals' social distance with immigrants while the degree of economic globalization plays little role. Although individuals' social distance with immigrants increases with right-wing populism, it decreases with social globalization. Our evaluation of this evidence is reliable because we also tested cross-level interaction effects and obtained a result that a greater degree of social globalization does not convey significant augmentation. This finding has important theoretical implications. Although the backlash of globalization is mostly due to economic globalization, aversion to social globalization

does not appear to be an issue in that the general public remains supportive of globalization in the EU and elsewhere [57]. Yet, our analysis raises a warning sign that people subject to a greater degree of social globalization will somewhat moderate the favorable influence of contact on attitudes toward immigrants. Nevertheless, in our evaluation of its magnitude, there is not sufficient evidence to assert the cultural globalization backlash argument [58]. This is especially notable given that immigrants to the EU are drawn mainly from the working and lower middle classes of developing countries [59].

Our study also resonates with recent research attention on the Russia-Ukraine war and COVID-19 pandemic. Unlike Syrian refugees in the 2015 crisis, Ukrainian refugees in 2022 have received warm welcomes throughout the EU, including countries where radical right parties are influential. The perception of Ukrainians as culturally and ethnically similar due to geographical proximity, shared histories, and cross-border connections are probably major factors, among others, accounting for differing attitudes [60, 61]. Alternatively, they are less likely to be seen as a distant outgroup as Syrians are in the EU. This testifies to the important implications generated from global exposure at the macro level, and of contacts at the micro level.

The current COVID-19 pandemic has made profound impacts on international migration with regards to its volume and flow. There is an opinion that the perceived threat of COVID-19 has lowered the acceptance of incoming immigrants. However, this does not tell the whole story. Recent research argues that there is no clear trend of increased antipathy toward immigration as a pandemic threat to health is considered a common threat and natives have adopted a prosocial attitude toward resident immigrants on the basis of a conception of "we are all in the same boat". This is indicative of increasingly identifying, rather than avoiding or detesting these outgroups in response to pathogen threats [62–64]. Despite evidence of the favorable response, there is still an issue around whether a higher threshold for immigrants approaching the border will be perceived as acceptable by the public. This is no doubt a legitimate inquiry for future research. However, none of the above-mentioned studies has taken exposure to global forces into consideration. It would be as important to investigate whether social globalization continues to play a crucial role in shaping attitudes and behavioral responses toward immigrants during and after the global pandemic.

We note some limitations and offer suggestions for future research. Firstly, global behaviors at the individual level, such as border-crossing, transnational networking, as well as the ability of speaking foreign languages [57], are not considered in this study. Their potential influences on perceived social distance with immigrants can be evaluated in order to gain a fuller understanding of the consequences of globalization at different levels. Secondly, we encourage researchers to pool cross-sectional data over time in order to observe changing attitudes toward immigration and examine the association of globalization with such attitudes from a within-country perspective. Thirdly, if populism measured as a political ideology of an individual is available [65, 66], it can replace the common measure of individual political ideology on the scale of right to left used in this study. Finally, comparable datasets outside the EU can be used to examine whether social distance is tied to contacts and globalization in similar or different manners across regions. This issue is worthy further examination for comparative researchers who pursue in-depth understanding of cross-national convergence or variation in response to immigration.

## Supporting information

**S1 Table. Country level variables for the 27 EU countries.**
(PDF)

## Acknowledgments

Rueyling Tzeng would like to thank Dr. Andrew Geddes, Professor and Director of the Migration Policy Center, European University Institute, Italy, and his colleagues for providing stimulating discussion in 2019 when she was a visiting fellow in the Center working on the draft of this co-authored manuscript.

## Author Contributions

**Conceptualization:** Ming-Chang Tsai, Rueyling Tzeng.

**Data curation:** Rueyling Tzeng.

**Formal analysis:** Ming-Chang Tsai.

**Investigation:** Ming-Chang Tsai, Rueyling Tzeng.

**Methodology:** Ming-Chang Tsai.

**Project administration:** Ming-Chang Tsai, Rueyling Tzeng.

**Resources:** Ming-Chang Tsai, Rueyling Tzeng.

**Software:** Ming-Chang Tsai.

**Supervision:** Ming-Chang Tsai, Rueyling Tzeng.

**Validation:** Ming-Chang Tsai, Rueyling Tzeng.

**Visualization:** Ming-Chang Tsai.

**Writing – original draft:** Ming-Chang Tsai, Rueyling Tzeng.

**Writing – review & editing:** Ming-Chang Tsai, Rueyling Tzeng.

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
