## [Decision Letter · Decision Letter 0]

30 Jun 2022

PONE-D-22-15355Globalization and social distance: Multilevel analysis of attitudes toward immigrants in the European UnionPLOS ONE

Dear Dr. Tzeng,

Thank you for submitting your manuscript to PLOS ONE. After careful consideration, we feel that it has merit but does not fully meet PLOS ONE’s publication criteria as it currently stands. Therefore, we invite you to submit a revised version of the manuscript that addresses the points raised during the review process.

You certainly have chosen a topical issue about the attitudes towards migrants in the European Union. To be able to consider the paper for publication in the journal, there is some substantial work to revise the manuscript. The two reviewers gave you very detailed critical remarks and comments (especially Reviewer 2), which need to be responded to, but first and foremost, considered deeply in your revision - as far as structuring your thoughts and presenting your findings, the methodology, additional literature, as well as conclusions are concerned. Take these points seriously and submit your corrected, more scientifically sound and elaborated paper.

We look forward to receiving your revised manuscript.

Kind regards,

István Tarrósy, PhD

Academic Editor

PLOS ONE

Journal Requirements:

Reviewers' comments:

Reviewer's Responses to Questions

**Comments to the Author**

1. Is the manuscript technically sound, and do the data support the conclusions?

Reviewer #1: Yes

Reviewer #2: Yes

2. Has the statistical analysis been performed appropriately and rigorously? 

Reviewer #1: Yes

Reviewer #2: Yes

3. Have the authors made all data underlying the findings in their manuscript fully available?

Reviewer #1: Yes

Reviewer #2: Yes

4. Is the manuscript presented in an intelligible fashion and written in standard English?

Reviewer #1: Yes

Reviewer #2: Yes

5. Review Comments to the Author

Reviewer #1: The paper is first and foremost based on an Eurobarometer survey from 2017. Bearing in mind events that took place since that the collection of that dataset, an attempt to use more up to date date should have been considered. The continuation (albeit in a more limited way) of the migration and asylum crisis, as well as the Covid-19 pandemic may have had a considerable effect on the examined issues. As an even more recent development, the Ukraine conflict's effects could also be considered. The secondary sources used should also be updated as a lack of recent literature is partly percievable. All in all the analysis definitely has merits, but the reader is left with a sense that had the paper been more up-to-date, it would have resulted in a much more valuable scientific contribution. In any case, the conclusion part should be considerably expanded as well.

Reviewer #2: Globalization and social distance: Multilevel analysis of attitudes toward immigrants in the European Union

Recently, migration, migration-related issues are undoubtedly significant research topics. The title promises that the authors adress their study to those migration patterns that affect the European Union. But at the same time authors have undertaken to find relation between globalization and social distance. As a reviewer I would suggest to make a much more narrow and clear focus in the abstract. In this shape the abstract is blurry and unclear to understand the real aim of this study.

Lately authors use EU and Europe as an interchangeable term, but it is not proper at all. If they speak about the EU, then authors ought to focus on the 26 member states formed and operated political and economic cooperation.

I would encourage authors to cite much more. Citations provide relevant info that writers are aware of the basic, related and adequate literature. This is now missing at many places, for instance in the 64-65th row. Just for sure, the EU did not receive "more than one million immigrants", but more than 1,5 million refugees / asylum seekers and economic migrants arrived and accrossed the external border of the EU, hundred thousands of them illegally, many of them as asylum seekers who applied for refugee status. The so-called hypotheses in the 67-68-69th sentence must be testified by using different methodology. First a theoretical and historical description what is totally missing now, and then the utilized multilevel analyses. What is also worth mentioning that seemingly the EU is a united entity, but in the reality it is not at all. There are 26 countries who are making different policies towards migration and migrants, seeking different solution for emerging challenges and attempt to find common and mutually beneficial solutions what is getting harder and harder as crisis phenomena multiply. The reviewer suggests the authors should use clear and more focused concepts in introduction also. There are a couple of authors who wrote about the EU-related migration and its implications, they should use their findings.

I think the most important finding can be found in sentences Nr. 102-103-104 for that is what the story is about. Not only the sentiments and attitudes should be detected and explained in the case of a given host society, but the perception also, what means, how a given society perceive the presence of migrants and what is the government policy towards immigrants and immigration. Furthermore, what is also worth mentioning the sentiment could change time after time as something unusual happen that might induce a shift in host society's attitude and triggers a persistent rejection of migrants OR against certain migration while others are still welcome. For understanding this very complicated topic the following study could help the authors: Segal, Uma A. 2019. Globalization, migration and ethnicity. Public Health 172:135-142. DOI: 10.1016/j.puhe.2019.04.011

The hyphotheses on page 9 and 10 are mostly facts and not presumptions and sometimes even not entirely true if we see for e. Japan. One of the engines of economic globalization but migration is still a very sensitive topic there.

I would also recommend authors should use updated surveys. Survey made in 2014 or 2012 might be outdated, mostly in the case of migration! Authors should find fresher surveys, that were conducted after 2015, the migration crisis.

At sentence Nr. 259 authors refer to Bogardus and social distance concept. They should introduce the definition social distance earlier, somewhere after the introduction.

On page 16 authors brought in new notions like populism, especially right-wing populism what is literally one of the consequences of the uncontrolled and non-regulated migration. But these parties rose their popularity by building politics on immigration and shortcomings of integration policies. I dont understand how this issue correlates with migration and social distance, because study did not investigate and unfold this topic properly. That's what I said before, the theme is too wide and that is why authors were not able to reveal aspects that would be relevant in this field. This problem appears again and again until the end of study.

I think discussion and conclusion part is the most useful in this study. Authors could summerized their thoughts in these pages. Unfortunately earlier their train of thought has been puzzled and confused. Authors should fix the abovementioned problems, revise given parts of their study and add more citations.

All in all the work would be a progressive study and bring new perspective into field of migration studies but authors could not realize their commitment entirely, because they wanted to express a lot of aspects but could present less than they expected.

6. PLOS authors have the option to publish the peer review history of their article (what does this mean?). If published, this will include your full peer review and any attached files.

Reviewer #1: No

Reviewer #2: No

---

## [Author Response · Author response to Decision Letter 0]

9 Aug 2022

Dear Academic Editor:

We are grateful for the comments and suggestions from you and the two anonymous reviewers for improving our paper. We have endeavored to incorporate these changes, rewritten numerous sections with additional literature, revised parts mentioned but misread by Reviewer 2, and extensively revised the concluding section following the reviewers’ advice. The current version is very much improved and satisfactory. Below we report our responses to all comments point-by-point.

Reviewer #1:

“…an attempt to use more up to date data should have been considered. The continuation (albeit in a more limited way) of the migration and asylum crisis, as well as the Covid-19 pandemic may have had a considerable effect on the examined issues. As an even more recent development, the Ukraine conflict's effects could also be considered. The secondary sources used should also be updated as a lack of recent literature is partly percievable…. In any case, the conclusion part should be considerably expanded as well.”

Answers: Our paper features a unique data set on social distance attitudes as a comprehensive measure, in contrast to a single response, of attitudes toward immigrants typically seen in other surveys. The data we used was collected in 2017. Indeed, this data set does not reflect changes due to the most recent events, as the reviewer correctly states, such as the Covid-19 pandemic and Ukraine conflict. We appreciated these suggestions and added further discussion in the concluding section to address the relevance of these critical global events although we have not found up-to-date data suitable for the research.

Reviewer #2: 

1. As a reviewer I would suggest to make a much more narrow and clear focus in the abstract. In this shape the abstract is blurry and unclear to understand the real aim of this study.

Answers: We have rewritten the abstract extensively and it now provides a better summary presentation of our study.

2. Lately authors use EU and Europe as an interchangeable term, but it is not proper at all. If they speak about the EU, then authors ought to focus on the 26 member states formed and operated political and economic cooperation.

Answers: In this paper, we used the term “European Union” (EU) most of the time and used “Europe” when referring the time before the establishment of the EU in 1993. Although the UK voted to exit the EU on June, 2016, this did not come into effect until January 31, 2020. Therefore, UK was included in our 2017 data set and thus we analyzed the 27 countries. In this revision, we added explanation for why UK was included so that readers will not misunderstand our use of the terminology concerning Europe and the EU. 

3. I would encourage authors to cite much more…

a. for instance in the 64-65th row. Just for sure, the EU did not receive "more than one million immigrants", but more than 1,5 million refugees / asylum seekers and economic migrants arrived and accrossed the external border of the EU, hundred thousands of them illegally, many of them as asylum seekers who applied for refugee status. 

Answers: We appreciate this suggestion and information. We have double checked and updated relevant statistics. According to the EU’s official statistics from Eurostat, the number of asylum seekers in 2015 European Refugee Crisis was over 1.2 million and the total number of long-term (at least a year) immigrants arriving in that year was more than 4.6 million. We included the information and sources in this revision. 

b. The so-called hypotheses in the 67-68-69th sentence must be testified by using different methodology. First a theoretical and historical description what is totally missing now, and then the utilized multilevel analyses. What is also worth mentioning that seemingly the EU is a united entity, but in the reality it is not at all.…There are a couple of authors who wrote about the EU-related migration and its implications, they should use their findings.

Answers: We stated our theoretical concerns in the 60th-62nd lines in the previous version (currently in the 66th-68th lines) “how the degree of globalization… solely or in combination with personal contact factors, influences native-born citizens’ attitudes toward immigrants”. As we have stated, this focal interest addresses a theoretical concern which has long been existent but understudied by way of empirical approaches. Admittedly, there are regional and country-specific historical backgrounds that deserve research attention, nevertheless, we have provided adequately historical background. In addition, following the suggestion, we have added a point in the introduction of this revision, with citations stating that the EU is not a homogenous entity and does not speak with one voice on migration policy, to strengthen our multilevel analyses, as this technique — a standard tool to assess both structural and individual level factors in pooled data from cross-national surveys — is employed to capture estimates of variance across countries. 

c. I think the most important finding can be found in sentences Nr. 102-103-104 for that is what the story is about. Not only the sentiments and attitudes should be detected and explained in the case of a given host society, but the perception also, what means, how a given society perceive the presence of migrants and what is the government policy towards immigrants and immigration. Furthermore, what is also worth mentioning the sentiment could change time after time as something unusual happen that might induce a shift in host society's attitude and triggers a persistent rejection of migrants OR against certain migration while others are still welcome. For understanding this very complicated topic the following study could help the authors: Segal, Uma A. 2019. Globalization, migration and ethnicity. Public Health 172:135-142. DOI: 10.1016/j.puhe.2019.04.011

Answers: Sentiment and perception probably are important parts of individual attitudes toward immigrants. We agree with the reviewer’s critical point of “how a given society perceive the presence of migrants and what is the government policy towards immigrants and immigration.” This is exactly what we have attempted in this study using a multilevel analysis to explain how certain populations perceived immigrants as close and acceptable, and therefore gave a warmer welcome than do others. As our dependent variable is an aspect of sentiment and perception, it is better not to include other aspects of sentiment and perception as independent variables lest an endogenous problem occurs (that is, potential fallacy in “attitudes affect attitudes” hypotheses). Nevertheless, this revision also incorporates Segal’s work (2019) in the introduction as suggested by the reviewer as it provides a useful overview of how increased protectionism across receiving countries has important policy implications for immigration. We also refer to this work in the concluding section.

4. The hypotheses on page 9 and 10 are mostly facts and not presumptions and sometimes even not entirely true if we see for e. Japan. One of the engines of economic globalization but migration is still a very sensitive topic there.

Answers: If our hypotheses are mostly facts, why are they not entirely true for Japan? The case of Japan can be an interesting reference as Japan has restrictive immigration policies and scores low in the globalization index. However, further discussion of this case might distract the readers so that we decided not to take the case of Japan into our paper. Additionally, it is beyond the scope of this study whose focus is the EU. 

5. I would also recommend authors should use updated surveys. Survey made in 2014 or 2012 might be outdated, mostly in the case of migration! Authors should find fresher surveys, that were conducted after 2015, the migration crisis.

Answers: We did not use any survey results of 2012 or 2014 in our research. There seems to be a misunderstanding in the reviewer’s reading. We investigated questionnaires about social distance used in other surveys and found them to be narrower in measurement. The questions about social distance in European Social Survey do not appear in every survey, and the latest one appeared in 2014. We briefly explain this in this revision. The latest version of the World Value Survey was conducted during 2017-2022 (wave 7), We have updated this part in this revision.

6. At sentence Nr. 259 authors refer to Bogardus and social distance concept. They should introduce the definition social distance earlier, somewhere after the introduction.

Answers: Thank you for this suggestion. We have moved it to the introduction in the 43nd-44th lines when we first stated the concept.

7. On page 16 authors brought in new notions like populism, especially right-wing populism what is literally one of the consequences of the uncontrolled and non-regulated migration. But these parties rose their popularity by building politics on immigration and shortcomings of integration policies. I don’t understand how this issue correlates with migration and social distance, because study did not investigate and unfold this topic properly. That's what I said before, the theme is too wide and that is why authors were not able to reveal aspects that would be relevant in this field. This problem appears again and again until the end of study…. authors could not realize their commitment entirely, because they wanted to express a lot of aspects but could present less than they expected.

Answers: In the 79th (currently 89th) line of page 4 in the Introduction section, we mention right-wing populism as one of our control variables selected from the most cited variables in the research on attitudes toward immigrants. As it is not the main interest of this research but can be a potential confounding factor, we offer only a limited explanation as we do for all other control variables in the page 16th in the measurement section. However, in this revision, we additionally cited previous research to explain that the support for such parties signifies a surge in anti-immigration sentiment. Having necessary controls in estimation has become a standard practice in empirical studies. In our view, inclusion of populism should not be considered lack of commitment. Rather, it can be seen as a necessary part of a rigorous procedure for reaching a decisive conclusion about whether globalization is a strong determinant.

8. I think discussion and conclusion part is the most useful in this study. Authors could summerized their thoughts in these pages. Unfortunately earlier their train of thought has been puzzled and confused. Authors should fix the abovementioned problems, revise given parts of their study and add more citations. 

All in all the work would be a progressive study and bring new perspective into field of migration studies but authors could not realize their commitment entirely, because they wanted to express a lot of aspects but could present less than they expected.

Answers: We are grateful for the suggestions of adding more citations and strengthening our conclusion, especially for the supportive evaluation that our study brings a progressive new perspective in the literature. We have added several relevant and updated citations. To strengthen our conclusion in comparative perspective and to bring new insight, we also added two new paragraphs to discuss the Ukrainian refugees (compared to Syrians) as well as the current covid-19 pandemic (lines 523-547 as follows).

“Our study also resonates with recent research attention on the Russia-Ukraine war and COVID-19 pandemic. Unlike Syrian refugees in the 2015 crisis, Ukrainian refugees in 2022 have received warm welcomes throughout the EU, including countries where radical right parties are influential. The perception of Ukrainians as culturally and ethnically similar due to geographical proximity, shared histories, and cross-border connections are probably major factors, among others, accounting for differing attitudes [61,62]. Alternatively, they are less likely to be seen as a distant outgroup as Syrians are in the EU. This testifies to the important implications generated from global exposure at the macro level, and of contacts at the micro level. 

The current COVID-19 pandemic has made profound impacts on international migration with regards to its volume and flow. There is an opinion that the perceived threat of COVID-19 has lowered the acceptance of incoming immigrants. However, this does not tell the whole story. Recent research argues that there is no clear trend of increased antipathy toward immigration as a pandemic threat to health is considered a common threat and natives have adopted a prosocial attitude toward resident immigrants on the basis of a conception of “we are all in the same boat”. This is indicative of increasingly identifying, rather than avoiding or detesting these outgroups in response to pathogen threats [63–65]. Despite evidence of the favorable response, there is still an issue around whether a higher threshold for immigrants approaching the border will be perceived as acceptable by the public. This is no doubt a legitimate inquiry for future research. However, none of the above-mentioned studies has taken exposure to global forces into consideration. It would be as important to investigate whether social globalization continues to play a crucial role in shaping attitudes and behavioral responses toward immigrants during and after the global pandemic.”

We again are appreciative of the constructive comments provided by you and the two reviewers. We are confident that this revised version is very much improved. 

Sincerely yours,

Ming-Chang Tsai

Research Fellow, Center for Asia-Pacific Area Studies

Deputy-Director, Research Center for Humanities and Social Sciences, Academia Sinica

and

Rueyling Tzeng (Corresponding author)

Research Fellow

Institute of European and American Studies

Academia Sinica

Taiwan

---

## [Decision Letter · Decision Letter 1]

8 Sep 2022

Globalization and social distance: Multilevel analysis of attitudes toward immigrants in the European Union

PONE-D-22-15355R1

Dear Dr. Tzeng,

We’re pleased to inform you that your manuscript has been judged scientifically suitable for publication and will be formally accepted for publication once it meets all outstanding technical requirements.

Kind regards,

István Tarrósy, PhD

Academic Editor

PLOS ONE

Additional Editor Comments (optional):

I can support the reviewers in their proposition to have your paper published in the journal, however, you still need to deal with the final minor issues indicated by Reviewer 2. The paper looks good and can add to the ongoing academic debate over the chosen topic.

Reviewers' comments:

Reviewer's Responses to Questions

**Comments to the Author**

1. If the authors have adequately addressed your comments raised in a previous round of review and you feel that this manuscript is now acceptable for publication, you may indicate that here to bypass the “Comments to the Author” section, enter your conflict of interest statement in the “Confidential to Editor” section, and submit your "Accept" recommendation.

Reviewer #1: All comments have been addressed

Reviewer #2: (No Response)

2. Is the manuscript technically sound, and do the data support the conclusions?

Reviewer #1: Yes

Reviewer #2: Yes

3. Has the statistical analysis been performed appropriately and rigorously? 

Reviewer #1: Yes

Reviewer #2: Yes

4. Have the authors made all data underlying the findings in their manuscript fully available?

Reviewer #1: Yes

Reviewer #2: Yes

5. Is the manuscript presented in an intelligible fashion and written in standard English?

Reviewer #1: Yes

Reviewer #2: Yes

6. Review Comments to the Author

Reviewer #1: (No Response)

Reviewer #2: Dear Author!

As the reviewer of the 1st version of your manuscript I have re-reviewed the new and corrected version what I found much more completed than the previous one. I appreciate that you used and inserted the recommended articles, materials and you took my advices.

The only thing I would clarify is the "misunderstanding" in the case of surveys. In this version as in the previous one you referred to surveys, here you are: "For example, the European Social Survey (2014) asks whether respondents would mind having an immigrant as their boss, or a close relative via marriage; the World Value Survey (2012) merely asked respondents whether or not they would like to have immigrants as neighbors." I said you might find more recent/current data on this topic. That was my suggestion, it was not misunderstanding at all... If it is possible, because both 2012, both 2014 have been before the migration crisis happened in 2015-2016.

Europe and European Union. Yes, you are right, the cooperation is called EU since 1993, before this year the cooperation was called European Economic Community (EEC) since 1957. As far as you know perfectly Europe is a continent, if you refer to this entity as a continent you are supposed to be right, but as a political-economic cooperation the adjective is wrong there. EU and EEC, these notions refer to Western Europe and the mass migration influenced this part of the European continent.

Good luck and congratulations for your work!

Kind regards,

7. PLOS authors have the option to publish the peer review history of their article (what does this mean?). If published, this will include your full peer review and any attached files.

Reviewer #1: No

Reviewer #2: No

---

## [Editor Report · Acceptance letter]

23 Sep 2022

PONE-D-22-15355R1 

Globalization and social distance: Multilevel analysis of attitudes toward immigrants in the European Union 

Dear Dr. Tzeng:

I'm pleased to inform you that your manuscript has been deemed suitable for publication in PLOS ONE. Congratulations! Your manuscript is now with our production department. 

Kind regards, 

on behalf of

Dr. István Tarrósy 

Academic Editor

PLOS ONE